# Full-Fat Soybean Meals as an Alternative Poultry Feed Ingredient—Feed Processing Methods and Utilization—Review and Perspective

**DOI:** 10.3390/ani14162366

**Published:** 2024-08-15

**Authors:** Ondulla T. Toomer, Edgar Orlando Oviedo-Rondón, Muhammad Ali, Michael Joseph, Thien Vu, Ben Fallen, Rouf Mian

**Affiliations:** 1Food Science & Market Quality & Handling Research Unit, Agricultural Research Service-United States Department of Agriculture, Raleigh, NC 27695, USA; thien.vu@usda.gov; 2Prestage Department of Poultry Science, North Carolina State University, Raleigh, NC 27695, USA; eooviedo@ncsu.edu (E.O.O.-R.); m.ali@uga.edu (M.A.); mvjoseph@ncsu.edu (M.J.); 3Soybean and Nitrogen Fixation Research Unit, Agricultural Research Service-United States Department of Agriculture, Raleigh, NC 27695, USA; ben.fallen@usda.gov (B.F.); rouf.mian@usda.gov (R.M.)

**Keywords:** poultry feed ingredients, soy and soybeans, feed processing

## Abstract

**Simple Summary:**

The U.S. Poultry industry utilizes approximately 67% of U.S. commercial defatted soybean meal annually. Feeding trials have demonstrated the effective utilization of soybean meal using various feed processing methods and feeding strategies. Nonetheless, few studies have examined processing methods and utilization of full-fat high-oleic soybeans as an alternative feedstock for poultry. This review aims to examine the current feed processing methods and utilization of conventional and full-fat high-oleic soybeans as preferable feedstock rations for poultry.

**Abstract:**

On a global scale, the poultry industry expands its wings in terms of meat and egg production to the masses. However, this industry itself requires a sustainable and permanent supply of different inputs, one of which is poultry feed and nutrition. Soybean is a versatile protein that is offered to poultry in different inclusion rates in commercial diets after being processed using various thermal and mechanical processing methods. Conventional commercial soybean meal is usually prepared by the extraction of oil from whole soybeans using solvents, producing a meal that has approximately 1% crude fat. Without oil extraction, full-fat soybean (FFSBM) is produced, and it is an excellent source of dietary energy and protein for poultry with a nutritional profile of 38–40% protein and 18–20% crude fat, on average. FFSBM has less crude protein (CP) than solvent-extracted soybean meal (SE SBM) but higher metabolizable energy due to higher fat content. Alternatively, extruded expeller processing produces defatted soybean meal containing approximately 6–7% crude fat. Studies have demonstrated that FFSBM can be used in poultry diets to improve poultry nutrition, performance, and quality of the poultry meat and eggs produced. This review aims to evaluate the nutrition and use of meals prepared from conventional and high-oleic soybeans using various feed processing methods.

## 1. Introduction

Soybeans originated in East Asia and were domesticated from wild soybeans (*Glycine soja*) between 1700 and 1100 B.C. and are now grown worldwide [1]. However, soybean production and utilization were limited within or near its center of Origin (the Orient) for centuries. The first recorded use of soybeans in the U.S. was in 1765 in Savannah, GA. It is thought that Samuel Bowen, a former seaman of the East India Company, brought soybeans to the U.S. from China via London. However, soybeans did not become widely cultivated in the U.S. until the mid-nineteenth century when soybean seeds were first grown in the Midwest. Soybeans continued to gain popularity in the 1870s when farmers began to use soybeans as forage for livestock [2,3], but it was not until 1917 that soybean meal was first reported for use as livestock feed. Since then, the soybean processing industry has continued to expand, and by the mid-1970s, the U.S. had been responsible for two-thirds of the world’s soybean production (University of Nebraska-Lincoln) [4]. Today, the U.S. is a global top producer of soybeans, with soybeans as the second largest row crop in the U.S. [5]. In 2023, 33.8 million hectares of soybeans were planted in the U.S., producing 4.16 billion bushels of soybeans. The total value of the crop exceeded $60.7 billion. From the 2023 soybean crop, the U.S. produced 48 million metric tons of soybean meal at an average price of $498.14 per metric ton. Also, an estimated 11.9 million metric tons of soybean oil was produced in the U.S. in 2023 at an average price of $1439 per metric ton [5].

Soybeans comprise almost 90% of U.S. oilseed production, with the U.S. as the second leading exporter of soybeans (USDA, ERS) [6]. More than 80% of U.S. soybeans are grown in the northern Midwest region of the U.S., with Iowa, Illinois, Minnesota, and Indiana as the top four soybean-producing states producing more than 49% of the annual U.S. supply [7]. Most U.S. soybeans are planted in late spring to early summer (May to June) and harvested in autumn (late September through October), depending upon the region (USDA, ERS) [8]. When mature in late September and October, the soybean seeds and plant turn yellow in color and are ready for harvest. With favorable dry weather conditions, the mature soybeans are harvested, dried, and cleaned from debris (sticks, stones, stems, etc.) and prepared for processing in the production of various meals for use in animal diets.

The solvent extraction process is used to produce SE SBM, and almost all oil is removed from soybeans. Without oil extraction, full-fat soybean meal (FFSBM) is produced, and it is a good source of dietary energy in poultry feed [9]. By using FFSBM, the cost of mixing liquid vegetable oil feed components can be avoided and is more economically feasible. Wider utilization of FFSBM has gained more acceptance, especially in regions without local soybean processing facilities. The use of FFSBM can save the cost of transportation associated with the shipment of soybeans to a large solvent extraction plant and return shipment. Hence, extrusion processing to produce FFSBM is easy to manage under these circumstances and is more practical and economically attractive [10].

The inclusion of FFSBM is an easy way to add fat to poultry diets without the need for special spraying devices in the mixer and liquid handling and storage. Generally, lipid feed components must be heated to a liquid state before adding to the feed, but this step can be omitted with the utilization of FFSBM. The lipid component of FFSBM is more resistant to lipid oxidation and rancidity and can be stored for a longer duration as compared to individually stored fat and oil feed components. FFSBM produced by organic soybeans is a great way to raise organic poultry for a niche market worldwide, and solvent-extracted soybean meal cannot be used for this purpose [11]. Interestingly, global utilization and market expansion of full-fat soybean meal is projected to have substantial growth between 2023 and 2031, driven by an 11.89% Compound Annual Growth Rate (CAGR), which is expected to increase the market value of soybeans to approximately $149.3 billion by the year 2031 [12].

Moreover, the COVID-19 global pandemic significantly impacted the production, supply, availability, and cost of palm (Malaysia) and coconut oil (Philippines), which are major sources of dietary energy in the diets of poultry and livestock globally and are utilized for binding in pelleted diets [13]. Reduced production and availability of these items have created substantial increases in the cost of these commodities and have been reflected in the rising global cost of livestock and poultry feed. Hence, it has become imperative to discuss the use of alternative feed ingredients that can serve as replacements for these vegetable oils in poultry feed formulations. In this review, we aim to revisit the use of full-fat soybean meal as a solution and alternative to these current global shortages and rising costs. While full-fat soybean meal prepared from conventional normal-oleic soybean cultivars has existed for over 50 years [13], utilization of full-fat soybean meal has been under-utilized within the poultry animal feed markets. Moreover, in this review, we explore the utilization of high-oleic soybean cultivars for the production of full-fat, high-oleic soybean meal is also explored. Thus, in this review we also aim to discuss the feed processing of normal-oleic and high-oleic soybeans with retention of their natural fat content and comparison of these soybean meals.

## 2. Nutrition

Soybeans have traditionally been grown primarily for their oil, with the remaining cake used to produce meal rich in digestible amino acids [14]. Ninety-five percent of the world’s extracted soy oil is used as edible oil in the food industry [15]. Interestingly, only a small percentage (2%) of the globally produced soybean meal is utilized for human consumption. Ninety-eight percent of globally produced soybean meal is predominately used for animal nutrition [16], with the poultry industry utilizing approximately 64%, swine-24%, beef and dairy cattle-10%, and 2% for domestic animals of all U.S. produced commercial soybean meal [17]. 

Soybeans provide more crude protein and a superior amino acid profile when compared to other legumes nutritionally [18]. Soybean meal (defatted, partially defatted) provides an ideal dietary amino acid profile when fed with corn in livestock rations, with the exception of methionine. Soybeans are limited in methionine; thus, finished feeds must be supplemented with synthetic methionine [18]. However, even with synthetic amino acid supplementation, performance can be adversely impacted by the presence of raw or marginally processed soybeans in finished feed [19]. Unprocessed soybeans and soybean meal are about 35% and 40% carbohydrates, respectively, with most present as non-structural oligosaccharides and a small portion as pectic polysaccharides [20], and approximately 6% as non-starch polysaccharide crude fiber [18] possessing anti-nutritional properties [21]. Unprocessed whole soybeans contain a number of anti-nutritional factors that have been shown to adversely affect animal growth performance caused by reduced protein digestion and absorption, resulting in reduced feed efficiency [22,23]. Among the most important anti-nutritional factors are trypsin inhibitors and lectins [23]. The two most crucial trypsin inhibitors found in unprocessed soybeans are Bowman-Birk inhibitors and Kunitz inhibitors, which provide protection to the germinating plant from microorganisms found within the soil and air prior to seed maturation. Typical levels of these trypsin inhibitors (TI) found in unprocessed soybeans range in activity levels between 20 and 35 g/kg, with the recommended threshold level of 4 g/kg in finished feed [17]. These trypsin inhibitors negatively impact nutrient digestion and absorption, resulting in reduced performance and animal growth. Animals fed diets containing unprocessed soybeans experience pancreatic enlargement as a result of overproduction of pancreatic trypsin and chymotrypsin [24,25]. Additionally, trypsin inhibitors have been shown to adversely impact nitrogen retention, causing increased excretion of metabolic nitrogen [26]. 

Soybean seeds and soybean meal contain a moderate level of carbohydrates (35–40%), with only a small fraction having anti-nutritional properties [21,25,27]. To date, heat treatment is the most used feed processing method to reduce and/or inactivate anti-nutritional factors. However, studies have shown that anti-nutritional factors, Bowman-Birk inhibitors, phytates, and oligosaccharides are not heat-labile [28]. Sixty to eighty percent of the total phosphorus found in unprocessed whole soybeans is phytate-bound and has reduced bioavailability due to the lack of endogenous phytase to digest phytate-bound minerals [29]. Moreover, studies have demonstrated that soybeans contain carbohydrase inhibitors, such as pectins and non-starch polysaccharides [21,27], which prevent the utilization and digestion of the carbohydrate fraction of the seed for dietary energy in poultry [28,30]. Unprocessed soybeans have moderate to high levels of trypsin inhibitors [23,25] in combination with low levels of sulfur amino acids, which greatly reduce the bioavailability of amino acids in soybeans [31].

Soybean breeders have worked for decades to identify quantitative trait loci (QTL) that are associated with phenotypic traits to enhance or alter seed nutritional composition or characteristics. Monsanto released the first genetically modified soybean line to the U.S. market with a glyphosate-resistant, “Roundup Ready” series of soybeans in 1996, and by 2014, 90.7 million hectares of these genetically modified soybeans were planted globally [32,33]. Methionine and cysteine content are traditionally limited in soybeans [34], and soybean breeders have been particularly interested in identifying QTL to enhance seed levels of methionine and cysteine and, hence, in soybean meal. Nonetheless, some studies have reported that genetically modified (GM) soybeans with increased total seed protein content had reduced crop yield and oil content [34,35]. 

Increased routine commercial use of affordable phytase feed supplementation has led to reduced emphasis on breeding programs aimed at reducing anti-nutritional factors in soybean lines [36,37,38]. Nonetheless, soybean breeding programs have successfully developed GM soybean lines with reduced levels of anti-nutritional factors [39]. In addition, soybean breeders have aimed to genetically enhance the levels of soluble sugars found in soybeans, such as sucrose (primary), fructose, and glucose (trace amounts), and multiple QTL have been identified that are associated with sucrose content in the seed [40,41]. Enhanced seed composition of sucrose and/or other sugars is highly desired for improved soy food flavor.

## 3. Processing Methods

“Cooking” is the simplest method for processing soybeans and other oilseeds. In these simple cooking methods, soybeans are submerged in water and heated for 30 minutes to 120 minutes, followed by air or mechanical drying. Other cooking methods involve roasting soybeans or other oilseeds using temperatures between 100 and 210 ℃, with various direct (flame) or indirect (coal burners, oven) heat sources with a minimum exposure time of approximately 20 seconds followed by milling. Flaking is a hydrothermal processing method that involves the use of injected low-pressure steam into a conditioner to cook the bean, releasing the oil of the beans in the press when rolled between two rollers, transforming the soybeans into flakes [42]. 

Processing of soybeans for oil production is conducted at soy oil processing plants, which traditionally extract the oil from the bean using solvent extraction methods. After harvest, the cleaned and dried soybeans are cracked under the pressure of corrugated rollers, creating bean particles of various sizes while removing the hull and outer husk (Figure 1). The removal of the outer husk prevents the retention of residual oil within the soy cake. Moreover, the removal of the outer husk reduces the fiber content while increasing the final protein content in the remaining soy cake after oil extraction [42]. Dehulled soybeans are conditioned using rotating drums with steam at 149 ºF. The conditioning and flaking process ruptures the cell walls of the seed, allowing the release of oil. During conditioning and flaking, the surface area of the soybean seed is increased by flattening and stretching the bean, causing the release of oil [42]. Lastly, successive hexane solvent washes are utilized to extract oil from the conditioned flakes. After solvent extraction, the residual hexane solvent is removed from the soybean flakes using de-solventization (Figure 1), producing the final soybean cake with the intended use as solvent-extracted defatted soybean meal [43,44]. 

Solvent-extracted soybean oil from conventional normal-oleic soybeans has a typical fatty acid composition of 51% linoleic acid, 7–10% α-linolenic acid, 23% oleic acid, 10% palmitic acid and 4% stearic saturated [45,46]. However, current plant breeding programs have modified the fatty acid profile of new soybean cultivars with genetic modifications to the Δ9 and Δ12 desaturases to enhance the oleic fatty acid content while keeping the linoleic levels low with reduced linoleic acid, producing high-oleic soybean cultivars with increased oxidative stability, positive sensory flavor profiles, and improved nutritional profiles for animal feed use [46]. In 2010, DuPont Pioneer released the first line of high-oleic soybeans with enhanced levels of oleic acid greater than 80% [46].

Oil extracted from high-oleic soybeans has a lipid fatty acid profile of about 75% oleic acid, 8% linoleic acid, 2% α-linolenic acid, and 12% saturated fats [47]. High-oleic soybean oil has superior heat stability and longer fry life compared to conventional soy oil. Moreover, studies have demonstrated that dietary replacement of vegetable oils high in saturated fatty acids with vegetable oils high in monounsaturated fats (oleic acid) may reduce the risk of coronary heart disease [48]. Hence, several animal feeding trials have been conducted to examine the dietary effects of the use of full-fat soybean meal and extruded soybean meal as an alternative to solvent-extracted defatted soybean meal and soy oil or poultry fat as separate feed ingredients in poultry feed formulations [49,50]. Additionally, these research interests have become of greater interest with the reduced competitive use of edible plant oils as biofuel [51]. 

Extruder-expeller processing is also utilized to produce partially defatted soybean meal [44], leaving 6–7% residual oil within the soy cake [42,43]. Dry extrusion processing uses no external heat and relies on the heat generated from the mechanical friction of the material passing through the screw barrel of the extruder. The mechanical screw press has a vertical feeder and a horizontal screw with an increasing diameter, which increases the pressure exerted on the oilseeds as they move along the length of the screw press. The barrel of the screw contains slots along the length, allowing the release of internal pressure and draining of extracted oil for collection in a trough below and the final discharge of the partially de-oiled cake at the end of the screw. Partially defatted soybean meal produced using extruder-expeller processing has traditionally been used in organic poultry feed, with nutritional qualities comparable to those of conventional solvent-extracted defatted soybean meal. Nonetheless, a number of investigations have reported large variations in the nutritional quality of extruded expeller-processed soybean meal with reduced amino acid composition [52]. 

Without oil extraction in the dry extrusion process, a full-fat soybean (FFSBM) meal is produced with a nutritional profile of approximately 38–40% crude protein and 18–20% crude fat [10,11,50]. Studies conducted by Ravindran et al. [53,54] demonstrated that FFSBM collected from four commercial feed mills in Southeast Asia had greater AME than defatted soybean meal (SBM) but less digestibility of dietary protein and amino acids, with values of crude protein, fat, AME, and standardized ileal digestibility coefficient of protein of 351 to 399 g/kg, 177 to 192 g/kg, 12.62 to 15.46 MJ/kg and 0.763 to 0.821, respectively. However, these studies utilized a small sample size and unknown processing history of the commercial feed mills selected. Chemical analysis of various processed soybean meals has shown that FFSBM has higher levels of metabolizable energy and crude fat, with partially defatted extruded expelled soybean meal and solvent-extracted defatted soybean meal providing higher levels of dietary protein [55,56]. 

Extrusion, micronizing, and jet-sploding are other processing methods used frequently with soybeans and other oilseeds. The extrusion process utilizes high temperatures (80–200 °C) with short intervals of time (10–270 s) for processing soybeans [57]. Soybean extrusion using a high-shear process requires a residence time of 15 seconds with a maximum temperature of about 5 seconds [57]. The extrusion process can be either performed using dry processing or wet processing. Dry processing occurs when frictional heat is produced between the beans and the extruder barrel as it travels along the barrel. Wet extrusion processing utilizes steam along with the mechanical energy generated in the screw press. Steam can also be utilized to precondition the beans or directly add them to the extruder barrel. The processing temperatures are different for both methods, with wet extrusion using lower temperatures (135–140 °C) [58,59], which is suitable in the presence of moisture/steam and hence requires less mechanical or frictional energy as compared to dry extrusion (150–160 °C) [59,60]. During dry extrusion, the pressure inside the extruder barrel is very high and can reach 40 atm [59]. Dry extrusion processing has greater screw speeds to generate the additional shear force required to properly process soybeans compared to wet extrusion. However, wet extrusion processing of preconditioned (steam) beans provides almost double the efficiency of the extruder and reduces the wear of the extruder barrel components by 20% [59]. The quality of soybean meal produced using different extrusion processing methods is related to the use of moisture during wet processing, which interferes with mechanical oil extraction. Upon comparison, FFSBM prepared using dry extrusion processing methods has a final moisture content of 5–6% as compared to 10% in FFSBM prepared by wet extrusion [60,61].

Micronizing is an infrared dry-heating processing method that uses radiant heat to thermally process grains [62]. Micronizing involves the use of electric burners or gas to heat ceramic plates that emit infrared dry radiant heat to the beans. This processing method is highly efficient and thermally penetrates beans. Jet-sploding is an alternative processing method for oilseeds that also reduces anti-nutritional factors found in unprocessed oilseeds. The jet-sploding process exposes the beans to a stream of pre-heated air between 140 and 315 °C, causing the heating of grain to occur from the inside out with an internal temperature of 90–95 °C, resulting in intracellular water boiling and cooking of the grains within. After the jet-sploding heating process, the beans are transferred into a cylinder mill to complete the process and allow the release of the intercellular oil [57,63]. 

## 4. Comparative Nutrition and Utilization

Initial efforts to include soybeans in animal diets were not fruitful due to growth retardation compared to those fed other protein sources. In 1917, Osborne and Mendel established a reasonable growth rate for livestock by adding heat treatment to whole soybeans prior to their inclusion in the diet [64]. Subsequently, it was observed that heat denatured protease inhibitors, which were responsible for growth retardation. Currently, feed manufacturers use conventional solvent-extracted defatted soybean meal as a standard against which other protein sources are generally compared [52].

Generally, soybeans contain 160–210 g/kg of oil, and upon solvent extraction of the oil, the remaining soya cake contains only 10 g/kg oil. In general, conventional normal-oleic soybean oil contains approximately 500 grams of unsaturated linoleic (18:2) fatty acid/kg as an indispensable fatty acid for poultry [55]. However, with the declining market for soy oil and increased consumer use of alternative plant-based oils, the use of full-fat soybeans in animal production has gained new interest in providing highly digestible protein and energy in the diets of poultry and other livestock. 

The polyunsaturated fatty acids (linolenic and linoleic) present in conventional FFSBM and soybean oil are highly labile to oxidation, causing rancidity, reduced shelf-life, and off-flavors in the oil and/or soy-oil-containing products. Consequently, genetic manipulation of conventional soybeans is creating high-oleic soybeans with improved lipid chemistry, elevated oleic fatty acid content, and extended shelf-life stability. Comparatively, the primary difference found in high-oleic soybean oil is the replacement of elevated linoleic acid levels by elevated oleic fatty acid levels and reduced linoleic and linolenic fatty acid levels (Table 1). Currently, the utilization of high-oleic soybeans has become a major research interest due to the increased stability and potential health benefits [48,65] of high-oleic FFSBM, high-oleic soy oil, and/or high-oleic soy oil-containing products. 

While the protein (37–38% crude protein) and lipid (16–18% crude fat) contents are similar between conventional normal-oleic and high-oleic soybean cultivars, the fatty acid profiles are dissimilar in the bean and the full-fat soybean meals that are produced (Table 2). The fatty acid profile of full-fat conventional soybean meal prepared from normal-oleic soybeans has a fatty acid profile of 11% palmitic acid, 4% stearic acid, 18% oleic acid, and 55% linoleic acid (Table 1). In comparison, the fatty acid profile of full-fat soybean meal prepared from high-oleic soybeans has a fatty acid profile of 7% palmitic acid, 3% stearic acid, 72% oleic acid, and 11% linoleic acid (Table 2). While full-fat soybean meal contains less crude protein as compared to defatted soybean meals, FFSBM contains greater digestible energy than defatted soybean meals comparatively (Table 2). Full-fat soybean meal delivers dietary fatty acids to poultry in excess of their nutrient requirements, which may adversely affect carcass quality with softer carcass fat and the potential for oxidation during storage. Hence, care must be taken when utilizing conventional full-fat soybean meal in broiler feed formulations [52].

FFSBM is a desirable high-energy poultry feed ingredient with enhanced AME in comparison to defatted soybean meal and may be alternatively utilized in food animal production as a replacement for these two-feed ingredients of defatted soybean meal and vegetable oil. However, it is necessary to consider the practical implications, economic viability, and inclusion rates. Experiments conducted by Mirghelenj et al. [69] revealed that wet extruded full-fat soybean meal (WE-FFSBM) can be used in broiler diets up to 15% (Table 3). In their experiment, full-fat soybean meal was wet extruded at a temperature of 155 °C for 15 s and added to four feed samples at 0, 7.5, 15, and 22.5% in place of dehulled soybean meal (D-SBM) and soybean oil. The growth performance of broiler birds was not influenced when WE-FFSBM was added up to 15%. However, when 22.5% WE-FFSBM was included in the diet, feed intake and body weight gains were reduced with reduced absorptive mucosal surface area in the upper small intestine of broilers. Due to a decrease in feed intake, body weight was reduced at the highest inclusion of FFSBM, which may be attributed to the presence of anti-nutritional factors in the FFSBM as a result of improper processing or heat treatment (Table 3). 

In parallel, Papadopoulos and Vandoros (1988) demonstrated that heat-treated FFSBM up to 15% in broiler diets did not adversely affect the body weights at 6 weeks of age [70]. Moreover, Wang et al. (2000) demonstrated that the growth performance of broilers fed a 15% FFSBM diet was equal to or better than that of control broilers fed a 15% defatted soybean meal diet supplemented with soy oil [71]. Todorov et al., (1999) reported no adverse effects on growth performance in broilers fed WE-FFSBM up to 20% of the finished diet [72]. Other studies have demonstrated that broiler growth performance was not affected at 21 and 42 days of age when WE-FFSBM was included in the diet up to 14% [73]. Another study reported that broiler body weights at 1 to 21 days of age were not affected when fed an FFSBM and solvent-extracted defatted soybean meal (SE SBM) blend [73]. However, body weights and feed-to-gain ratios were improved in broilers fed an FFSBM:SE SBM blend of 75/25 and 100/0 at 1 to 42 days (Table 4). 

Erdaw et al. (2017) conducted experiments to study the growth and physiological response of broilers fed diets containing raw FFSBM supplemented with high-impact microbial protease [27]. Raw FFSBM was added at 0, 10, and 20% to replace defatted soybean meal, and three levels of mono-component protease, derived from *Nocardiopsis prasina* were added at 0.1, 0.2 and 0.3 g/kg diet (equivalent to ~7500, 15,000, and ~22,500 protease units/kg diet, respectively) for 0–35 days. By increasing the raw FFSBM alone in the diets, there was an increase in the weight of the pancreas by 24, 32, and 26% at days 10, 24, and 35 (Table 5), respectively, but a decrease in apparent ileal digestibility of crude protein and amino acids at day 24. The highest villus length and mucosal depth were observed compared to other time points (day 24) when 20% raw FFSBM was supplemented with 0.2 g protease/kg. It is important to note that when fed in combination, raw FFSBM and protease supplementation did not significantly alter the body weight of broilers in comparison to the controls during the entire period (0–35 days).

In laying hen diets, FFSBM is also a valuable source of dietary protein and high-quality fats. Also, the use of FFSBM is often a preferable feed ingredient to vegetable oil sources, particularly on farms and feed mills that lack liquid handling capabilities. FFSBM provides ample amounts of lysine, the profile of unsaturated fatty acids, vitamin E, and linoleic acid [74,75]. In addition to these nutrients, FFSBM provides almost 30% to 50% of the nutritional requirements of riboflavin, niacin, pantothenic acid, thiamine, folic acid, and choline found within the lecithin of whole soybeans, all nutrients vital for egg production in egg-producing laying hens [76]. Additionally, the use of FFSBM may be preferable due to less dustiness and improved palatability of the feed of laying hen diets [75,76]. The inclusion of 11% of a full-fat extruded soybean meal diet of laying hens reduced the cholesterol content from 11.3 to 10.1 mg/g in egg yolk, with increased egg production from 275 eggs to 283 eggs, egg weight from 60.3 to 61.6 g and FCR of 2.29 during the lay period of 324 days which was unchanged in comparison of the control group [76]. Feed intake was also noticed to have increased from 117.1 g to 122.9 g [75]. Senkoylu et al., (2005) conducted experimentation by incorporating 10, 16, and 22% FFSBM in layer diets in comparison to a conventional control diet during post-peak egg production [75]. Experimental diets were provided to hens (Bovans White Strain) from 33 to 42 weeks. In this study, it was observed that FFSBM can be added up to 22% in laying hens’ diet without any adverse effects and with improvement in egg mass and FCR [75]. Egg production was not affected so much between the control and FFSBM treatment groups. However, feed intake was reduced at FFSBM inclusion levels of 10% and 16% in comparison to the controls. In this study, egg weights (g/egg) were not significantly altered, yet there was an increase of almost 1 g between inclusion levels of 10% and 22% of FFSBM. Furthermore, egg mass (g/hen/day) and FCR were improved with the increasing inclusion of FFSBM (Table 6).

Another aspect of this study covers the effect of different dietary inclusion levels of FFSBM on internal and external egg qualities, both of which were not adversely affected in comparison to conventional eggs [76]. In addition, shell weight and shell thickness were not significantly different between eggs produced from laying hens fed diets with FFSBM and those fed conventional controls [76]. The same was the case for shell thickness, Haugh unit, checked, and percentage of cracked eggs. (Table 6). Interestingly, the most recent nutrient digestibility studies conducted in layer hens fed diets containing either full-fat (normal-oleic and high-oleic) or defatted (solvent extracted and extruded expelled) soybean meals (Maharjan et al., 2023) reported similar crude fat digestibility values ranging from 71 to 84 % and crude protein digestibility values ranging from 67 to72 % (*p* > 0.05) [49].

Ali et al., (2024) [50] conducted a broiler feeding trial comparing a blend of SE SBM with either extruded expelled normal-oleic defatted soybean meal (EENO), full-fat normal-oleic soybean meal (FFNO), or full-fat high-oleic soybean meal (FFHO). Five hundred forty Ross-708 male broilers were randomly assigned to three isocaloric, isonitrogenous dietary treatment groups for 6 weeks (Table 7). The EENO blend consisted of 5 to 7% SE SBM and 25 to 32% extruder expelled normal-oleic defatted soybean meal. The FFNO blend consisted of 5 to 17% SE SBM and 25 to 31% full-fat normal-oleic soybean meal. The FFHO blend consisted of 5 to 17% SE SBM+25 to 31% full-fat high-oleic soybean meal. Broilers fed diets containing a blend of SE SBM and either full-fat normal-oleic or high-oleic soybean meals had acceptable performance parameters compared to broilers fed an extruded expelled defatted soybean meal diet [50]. From days 0 to 14, broilers fed the EENO had greater body weights as compared to broilers fed the full-fat normal-oleic (FFNO) and full-fat high-oleic (FFHO) soybean meal diets (*p* < 0.001) [77]. However, there were no significant differences in the feed conversion ratio between the treatment groups from 0 to 14 days. Nonetheless, there were no significant differences in the final 0-to-47-day body weights or feed intake (FI) between the treatment groups (*p* > 0.05), while FCR was best (lowest value) in birds fed the EENO diet as compared to the FFHO treatment group with similar FCR values between the EENO and FFNO treatments. Also, broilers fed a blend of SE SBM and full-fat high-oleic soybean meal (FFHO) produced poultry meat (*Pectoralis major*) that was enriched with oleic acid (*p* < 0.001) and reduced saturated fat (palmitic, stearic and lignoceric acid) content (*p* < 0.01) as compared to poultry meat produced from broilers fed a diet containing extruded expelled defatted soybean meal diet or full-fat normal-oleic soybean meal (Table 8).

Comparatively, layer performance (final average body weights, feed conversion ratio, number of eggs produced, and average weekly egg mass-Table 9) was dissimilar between hens fed diets containing either 21% solvent-extracted defatted soybean meal (SE SBM), 21% extruded expelled defatted normal-oleic soybean meal (EENO), 25% full-fat normal-oleic soybean meal (FFNO), or 24% full-fat high-oleic soybean meal (FFHO) in a 3-week feeding trial [49]. In parallel to the findings of Ali et al. (2024) [50], eggs produced from hens fed a full-fat, high-oleic soybean meal diet were enriched with oleic acid (*p* < 0.0001) and reduced saturated fat (palmitic and stearic acid) content (*p* < 0.0001) as compared to eggs produced from hens fed the other dietary treatment groups (Table 10). Eggs produced from hens fed the FFHO dietary treatment also had linoleic and linolenic acid contents that were significantly lower as compared to eggs produced from hens fed the FFNO treatment (*p* < 0.0001) [49]. Taken together, the utilization of high-oleic soybeans and/or full-fat high-oleic soybean meal has become a major research interest due to the increased shelf stability of high-oleic soybean-containing products and due to the potential health benefits in animal production nutrition and health and/or improved lipid chemistry of the meat, eggs or milk produced.

## 5. Conclusions

Full-fat soybeans are a valued feed ingredient for utilization in poultry rations, with increasing commercial utilization as a blend of high-quality protein and oil with less variability in composition. The benefits of FFSBM and soybeans may be realized with optimal processing to adequately denature and reduce anti-nutritional factors while sparing amino acid and protein content, quality, and bioavailability. This overview of studies demonstrates that optimal extrusion processing needed to inactivate heat labile anti-nutritional factors produces FFSBM with a targeted urease index of 0.20 to 0.05 Ph unit change along with 70–85% of KOH solubility. 

Furthermore, this overview of poultry feeding trials demonstrates that inclusion levels of up to 31% in the diets of broilers and up to 25% in the diets of laying hens of full-fat soybean meal do not adversely affect growth, performance, or survivability. Even so, much research remains to be conducted on the effective utilization and processing methods of full-fat, high-oleic soybean meal and/or soybeans in animal food production.

## 6. Perspectives

Soybeans (Glycine max) are the most valuable and widely grown oilseed globally, with a 350% increase in global crop production between 1987 and 2018 [68]. Today, the U.S., Brazil, and Argentina are the largest global producers of soybeans, while China and other Asian countries are the largest global consumers of soybeans, with U.S. exports of soybeans to China totaling over $3 billion in 2019 [77]. Much of the early success of soybeans is attributed to the U.S. Department of Agriculture (USDA) and commodity-based organizations. The Office of Foreign Seed and Plant Introduction was established in 1898 by the USDA and was responsible for introducing new soybean germplasm. Prior to 1898, only eight soybean varieties were reported to have been grown in the U.S [66]. In 1929, two USDA scientists, Howard Dorsett and William Morse, went to Japan, Korea, and China to collect soybean germplasm, which totaled over 4000 soybean accessions over the course of two years. Currently, there are more than 22,000 soybean lines in the USDA Soybean Germplasm Collection in Urbana, IL [67,78].

The establishment of a soybean checkoff fund approved by Congress in 1990–1991 (Soybean Promotion, Research and Consumer Act) with contributions of 0.5% of the price of each bushel sold by soybean farmers was appropriated for sustained investments in research (in breeding, production, abiotic and abiotic resistance, new uses), and marketing has been the key to the fast and wonderful success of soybeans in the U.S. Furthermore, this success has been spear-headed under the governance of the United Soybean Board, which directs the soybean checkoff’s national efforts. The United Soybean Board consists of 78 volunteer farmer directors, often nominated by their state-level soybean checkoff organizations and appointed to the national board by the U.S. Secretary of Agriculture [79]. In addition, many states have local soybean boards that invest local checkoff funds into their respective state programs.

The establishment of the soybean checkoff fund and the United Soybean Board, along with state checkoff funds to support sustained investments in research and marketing, catapulted the fast and wonderful success stories of U.S. soybeans. As the economies of developing countries continue to improve, the global population’s shifting dietary preferences favoring animal proteins have resulted in dramatic increases in the demand for poultry, livestock, and fish around the world, in turn increasing the demand for soybean feed products. Sustained investment by the soybean checkoff fund, alongside further improvement of world economies, will result in further expansion and demand for soybean products in the future. With continually growing economic development globally, and growing consumer demand for high-quality poultry meat and eggs, and soy-containing foods, global soybean sectors are projected to expand with greater market utilization within animal food production and food sectors and the subject of much discussion.

In the last decade, USDA scientists have aimed to conduct “farm to fork” applied research to meet USDA research initiatives to deliver a 20% increase in the quality of agricultural commodities with a 20% reduction in environmental impact and to improve the nutritional quality of the foods produced. As such, much research, development, and investment has been focused on the development of improved seed cultivar varieties and technological applications to enhance the quality of agricultural commodities and products for use in poultry and animal feed and food. 

While the use of conventional full-fat soybean meal offers the benefits of reduced need for feed processing methods and cost association with solvent extraction and the need for liquid storage and handling of vegetable oils at the feed mill, utilization of high-oleic full-fat soybean meal offers a plethora of additional end-user benefits. Utilization of high-oleic full-fat soybean meals as a poultry feed ingredient reduces lipid oxidation and rancidity in finished poultry feed products and nutritionally enriches poultry meat and eggs produced with monounsaturated oleic acid, with the potential for human nutrition and health benefits. Given this, it is expected that the high-oleic soybean global market and utilization will greatly expand, driven by increased consumer preferences and demand for healthier, high-quality foods produced sustainably. The high-oleic soybean market is expected to have significant growth, with a projected CAGR of 12.8% over the next seven years, driven by the demand for healthier vegetable cooking oils and increasing awareness regarding the health benefits of consuming high-oleic soybean products. 

Due to this ever-changing global landscape directly impacting the global supply, exchange, production, and cost of agricultural commodities, it is imperative that we examine the current and projected utilization of alternative feed ingredients for use in poultry and animal feed formulations. The global COVID-19 pandemic changed the trajectory of the production, supply, and cost of a number of conventional poultry feed ingredients, such as solvent-extracted soybean meal and palm and coconut vegetable oils. Hence, in this manuscript, we discuss poultry research with the replacement of these ingredients with the use of full-fat normal-oleic or high-oleic soybean meals in poultry diets.

High-oleic soybeans and full-fat high-oleic soybean meal may be a feed for the global future due to improvements in lipid chemistry and benefits offered within poultry, animal, and human nutrition. Nonetheless, much additional applied research is needed to adequately assess the limitation of high-oleic full-fat soybean meal as an alternative poultry and animal feed ingredient. 

## Figures and Tables

**Figure 1 animals-14-02366-f001:**
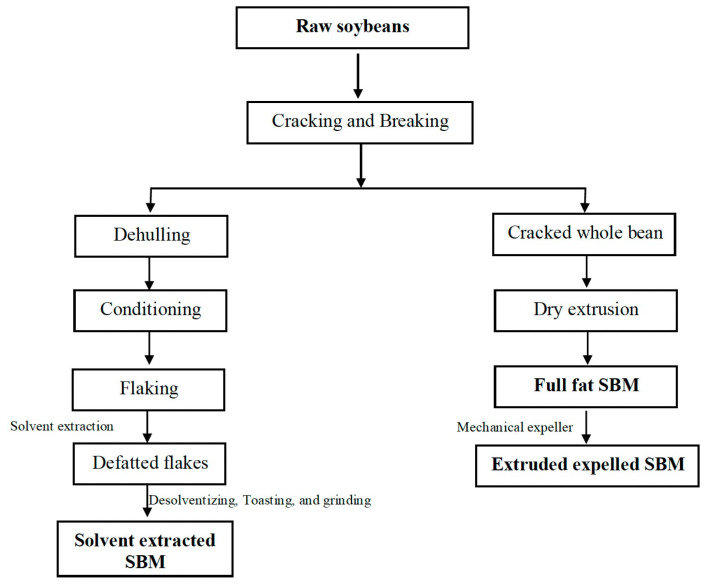
Soybean Processing Using Solvent Extraction and Extrusion Processing. Raw soybeans dried using ambient air to less than 12% moisture required for the processing and production solvent-extracted SBM or full-fat SBM. A mechanical expeller further processes full-fat SBM to produce extruded expelled SBM [44].

**Table 1 animals-14-02366-t001:** Fatty Acid Profile of Conventional and High-Oleic Soybean Oil *.

Major Fatty Acid	Conventional Soybean Oil (%)	High-Oleic Soybean Oil (%)
C 16:0 (palmitic)	10	7.3
C 18:0 (Stearic)	4	3.4
C 18:1 (Oleic)	18	76–85
C 18:2 (Linoleic)	55	1.3–6.7
C 18:3 (Linolenic)	13	1.6–2.0

* [66] Warner and Gupta; [67] Napolitano et al. Comparative fatty acid profile of the lipid composition of conventional normal-oleic and high-oleic soybean oils extracted from whole soybeans. Each fatty acid represents a percentage of the total lipid content.

**Table 2 animals-14-02366-t002:** Comparative nutrient compositions of soybeans and soybean meals *.

	NO-Soybeans	HO-Soybeans	SE-SBM	FF-SBM	FF-HO SBM
CP (%)	36.6	38.2	48.0	38.0	39.9
CF (%)	17.9	16.4	4.75	17.3	15.5
GE (kcal/kg)	5235	5236	4105	4863	4890
AME-(kcal/kg)	-	-	2614	3637	3650
Palmitic Acid (%)	10.5	6.92	14.1	11.1	7.74
Stearic Acid (%)	2.89	0.58	3.67	3.55	3.11
Oleic Acid (%)	19.5	81.5	14.3	18.0	71.7
Linoleic Acid (%)	51.6	4.76	55.9	55.2	11.0

* [68] Willis. Values are representative on an as-is basis. Each fatty acid represents a percentage of the total lipid content. Abbreviations: NO-soybeans (conventional normal-oleic soybeans); HO-soybeans (high-oleic soybeans); SE SBM (solvent-extracted defatted soybean meal); FF-SBM (full-fat soybean meal); CP (crude protein); CF (crude fat); GE (gross energy).

**Table 3 animals-14-02366-t003:** Effect of dietary inclusion rate of full-fat soybean on feed intake, weight gain, and FCR of Broiler Chickens *.

	FI (g/bird/d)	Weight Gain (g/bird/d)	FCR (g:g)
Level of FFSBM	Age (0 to 42 d)		
0	86.83 ^a^	46.7 ^a^	1.86
7.5	84.20 ^ab^	44.2 ^ab^	1.90
15	83.88 ^ab^	45.2 ^ab^	1.84
22.5	81.59 ^b^	43.3 ^b^	1.88
SEM	0.89	0.39	0.04
*p*-value	0.04	0.011	0.77

* [69] Mirghelenj et al. Results shown here represent the data for the whole experimental period (0–42 d) for Ross 308 male broiler chicks. Values are the average of four replicates of 12 birds per diet (48 birds per diet). Abbreviations: FI (feed intake); FCR (feed conversion ratio). ^a,b^ Means that do not share superscript letters in a column are significantly different (*p* < 0.05).

**Table 4 animals-14-02366-t004:** Effect of dietary inclusion rate of full-fat soybean on feed intake, weight gain, and FCR of Broiler Chickens *.

FFSBM: SE-SBM Ratio	Body Weights		Feed:Gain Ratio	
	21 day	42 day	21 day	42 day
0:100	0.762	2.534 ^c^	1.290 ^a^	1.670 ^a^
25:75	0.772	2.520 ^c^	1.283 ^a^	1.660 ^ab^
50:50	0.784	2.539 ^c^	1.239 ^b^	1.631 ^bc^
75:25	0.788	2.583 ^b^	1.243 ^b^	1.610 ^cd^
100:0	0.774	2.594 ^a^	1.229 ^b^	1.582 ^d^

* [73] Subuh et al. Results shown here represent the data for the whole experimental period (0–42 d) for Cobb 500 male broiler chicks. Values are the average of six replicates of 60 birds per treatment. ^a–d^ Means that do not share superscript letters in a column are significantly different (*p* < 0.05).

**Table 5 animals-14-02366-t005:** Main Effects of Raw Soybean Meal and Protease Supplementation on Broiler Chicken Live Body Weight (g/bird) and Weight of Pancreas (g/100 g body weight) *.

	Live Body Weight (g)			Weight of Pancreas (g/100g BW)		
Items	10 d	24 d	35 d	10 d	24 d	35 d
RSBM (%)						
0	282 ^a^	1463	2540	0.54 ^c^	0.23 ^c^	0.15 ^b^
10	279 ^a^	1420	2432	0.61 ^b^	0.31 ^b^	0.20 ^a^
20	265 ^b^	1419	2464	0.71 ^a^	0.34 ^a^	0.21 ^a^
Protease (g/kg)						
0.1	270 ^b^	1424 ^ab^	2422	0.62	0.29	0.19
0.2	275 ^ab^	1415 ^b^	2461	0.61	0.29	0.19
0.3	282 ^a^	1463 ^a^	2549	0.64	0.29	0.18
Sources of Variation						
RSBM	0.001	0.123	0.10	0.001	0.001	0.01
Protease	0.005	0.03	0.06	0.102	0.612	0.423
RSBM × Protease	0.511	0.432	0.712	0.231	0.641	0.451

* [27] Erdaw et al. Values are the average of 162 birds (male Ross 308) per treatment. Protease = mono-component protease derived from *Nocardiopsis prasina* (Ronozyme ProAct) (DSM Nutritional Products, Australia Pty. Ltd, Wagga Wagga, NSW, Australia), was added at 0.1, 0.2, or 0.3 g/kg, equivalent to ~7500, 15,000, and ~22,500 protease units/kg diet. Abbreviations: RSBM (raw soybean meal). ^a–c^ Means that do not share superscript letters in a column are significantly different (*p* < 0.05).

**Table 6 animals-14-02366-t006:** Effects of Full-fat Soybean Meal Levels on Bovans White Hen Performance and Egg Qualities (33 to 42 weeks of age) *.

	FFSBM Inclusion, %				SEM	*p*-Value
	0	10	16	22		
Performance data						
Hen day production	96.7	96.1	97.2	97.3	0.285	0.443
FI, g/hen/day	111.7 ^a^	105.6 ^c^	108.3 ^bc^	109.5 ^ab^	0.610	0.002
Egg weight, g	61.8 ^ab^	61.1 ^b^	61.8 ^ab^	62.2 ^a^	0.136	0.038
Egg mass, g/hen/ day	59.7 ^ab^	58.7 ^b^	60.1 ^a^	60.5 ^a^	0.205	0.010
FCR, g feed:g egg	1.870 ^a^	1.799 ^b^	1.804 ^b^	1.809^b^	0.010	0.026
External egg qualities						
Shell weight, g	7.5	7.6	7.4	7.5	0.055	0.828
Shell weight, %	12.0	12.5	12.3	12.3	0.079	0.230
Shell thickness, μ	294	298	297	296	1.449	0.741
Checked and cracked eggs, %	0.360	0.326	0.359	0.236	0.069	0.919
Internal egg qualities						
Albumen height, mm	6.07	5.83	6.29	6.18	0.101	0.424
Haugh units	83.3	81.8	84.0	84.2	0.596	0.479

* [76] Senkoylu et al. The total number of treatments was 4, and the values were the average of 6 replicates per treatment (total replicates = 24) and 12 birds in each for 72 birds per treatment (Bovans White strain hens). FFSBM was substituted at the expense of soybean meal and soybean oil. Abbreviation: FFSBM (full-fat soybean); FI (feed intake); FCR (feed conversion ratio). ^a–c^ Means that do not share superscript letters in a column are significantly different (*p* < 0.05).

**Table 7 animals-14-02366-t007:** Effects of conventional or full-fat high-oleic soybean meal on live performance of Ross 708 Broiler Chickens *.

0–14 days (*n* = 40)			
Treatment	BW (g)	FI (g)	Adj FCR (g/g)
EENO	526 ^a^	558 ^a^	1.146
FFNO	497 ^b^	529 ^b^	1.155
FFHO	508 ^b^	541 ^ab^	1.165
SEM	4	6	0.007
*p*-values	<0.001	0.008	0.150
0–47 days (*n* = 40)			
Treatment	BW (g)	FI (g)	Adj FCR (g/g)
NO-EE SB	3603	5568	1.532 ^b^
NO-FF SB	3522	5416	1.537 ^ab^
HO-FF SB	3545	56 77	1.603 ^a^
SEM	36.03	89.37	0.019
*p*-values	0.291	0.136	0.026

* [50] Ali et al. Five hundred forty Ross-708 male broilers (18 broilers/pen, 10 replicates/treatment) were randomly assigned to three isocaloric, isonitrogenous dietary treatment groups. The starter diet (22.8% crude protein, 3000 kcal/kg) was fed from to 0–14 days, grower diet (20.7% crude protein, 3100 kcal/kg) was fed from 15 to 35 days, and finisher diet (19.0% crude protein, 3200 kcal/kg) was fed from 36 to 42 days. Body and feed weights were collected weekly for 6 weeks. Dietary Treatments: EENO-blend of 5 to 7% SE SBM + 25 to 32% extruder expelled normal-oleic defatted soybean meal, FFNO-blend of 5 to 17% SE SBM + 25 to 31% full-fat normal-oleic soybean meal, FFHO-blend of 5 to 17% SE SBM + 25 to 31% full-fat high-oleic soybean meal. Abbreviations: BW-body weight; FI-feed intake; Adj FCR- adjusted feed conversion ratio adjusted by mortality weights. ^a,b^ Means that do not share superscript letters in a column are significantly different (*p <* 0.05) by Tukey’s test.

**Table 8 animals-14-02366-t008:** Comparative fatty acid profile of breast meat produced from broilers fed a full-fat soybean meal diet *.

Fatty Acids %	Pectoralis Major			SEM	*p*-Value
	EENO	FFNO	FFHO		
Oleic acid (18:1, cis)	31.50 ^b^	26.57 ^c^	48.85 ^a^	0.50	<0.001
Linoleic acid (18:2, cis)	25.66 ^b^	32.61 ^a^	14.87 ^c^	0.33	<0.001
Palmitic acid (16:0)	19.83 ^a^	17.95 ^b^	17.07 ^c^	0.19	<0.001
Stearic acid (18:0)	6.69 ^a^	6.73 ^a^	5.58 ^b^	0.11	<0.001
Elaidic acid (18:1, trans)	0.07 ^a^	0.02 ^ab^	0.00 ^b^	0.01	0.008
Linolenic acid (18:3, n6)	0.26 ^a^	0.24 ^a^	0.20 ^b^	0.01	<0.001
Gondoic acid (20:1)	0.28 ^b^	0.24 ^c^	0.42 ^a^	0.01	<0.001
Lignoceric acid (24:0)	0.92 ^a^	0.85 ^a^	0.67 ^b^	0.05	0.01

* [50] Ali et al. Five hundred forty Ross-708 male broilers (18 broilers/pen, 10 replicates/treatment) were randomly assigned to 3 isocaloric, isonitrogenous dietary treatment groups. The starter diet (22.8% crude protein, 3000 kcal/kg) was fed from 0 to 14 days, grower diet (20.7% crude protein, 3100 kcal/kg) was fed from 15 to 35 days, and finisher diet (19.0% crude protein, 3200 kcal/kg) was fed from 36 to 42 days. Dietary Treatments: EENO-blend of 5 to 7% SE SBM + 25 to 32% extruder expelled normal-oleic defatted soybean meal, FFNO-blend of 5 to 17% SE SBM + 25 to 31% full-fat normal-oleic soybean meal, FFHO-blend of 5 to 17% SE SBM+25 to 31% full-fat high-oleic soybean meal. At termination (6 weeks), four broilers per pen were processed and breast meat samples were collected and evaluated for fatty acid analysis using standard AOAC-approved procedures by a commercial laboratory (ATC Scientific, Little Rock, AR, USA). ^a–c^ Means that do not share superscript letters in a column are significantly different (*p <* 0.05) according to Tukey’s test.

**Table 9 animals-14-02366-t009:** Effects of conventional or high-oleic full-fat soybean meal on layer performance *.

Treatment	BW (g)	FCR	Egg Production	Egg Mass (g)
Control	1693	1.727	92.71	63.16
EENO	1628	1.601	96.35	60.20
FFNO	1660	1.660	93.75	60.98
FFHO	1695	1.641	94.27	62.52
SEM	45.9	0.046	2.292	1.383
*p*-value	0.417	0.065	0.451	0.136

* [50] Maharjan et al. White Shaver hens at the peak of production (≈ 40 weeks of lay) were individually housed and randomly assigned to one of four isocaloric (2,927 kcal/kg), isonitrogenous (18.5% crude protein) dietary treatments (12 hens per treatment) and provided feed and water freely for approximately 3 weeks. Treatments were the following: Control = conventional diet containing solvent-extracted defatted soybean meal and corn; EENO = diet containing extruded expelled defatted normal-oleic soybean meal and corn; FFNO = diet containing full-fat normal-oleic soybean meal and corn; FFHO = diet containing full-fat high-oleic soybean meal and corn. Feed intake was calculated weekly. Individual final body weights (BW) were collected at termination (3 weeks), and eggs were collected daily. Eggs were weighed and enumerated weekly. FCR = total egg mass (g)/total feed consumed (g). Egg production was calculated as the total number of eggs produced over 3 weeks divided by the total number of hens. Egg mass = weekly average egg weights per treatment.

**Table 10 animals-14-02366-t010:** Fatty Acid Profile of Eggs Produced from Hens Fed a Full-fat High-Oleic Soybean Meal Diet *.

	Treatments					
	Control Diet	EENO Diet	FFNO Diet	FFHO Diet	SEM	*p*-Value
Crude Fat, % ^2^	34.0	33.3	33.1	33.5	0.35	0.69
Palmitic %, (C16:0)	24.2 ^a^	23.3 ^b^	23.2 ^b^	22.1 ^c^	0.14	<0.0001
Stearic %, (C18:0)	9.8 ^a^	9.7 ^a^	10.1 ^a^	7.6 ^b^	0.19	<0.0001
Oleic %, (C18:1)	36.0 ^b^	35.2 ^bc^	34.8 ^c^	50.7 ^a^	0.28	<0.0001
Linoleic %, (C18:2)	21.0 ^c^	22.5 ^b^	23.3 ^a^	11.3 ^d^	0.24	<0.0001
Linolenic %, (C18:3)	0.94 ^b^	1.14 ^a^	1.11 ^a^	0.43 ^c^	0.35	<0.0001

* [49] Maharjan et al. Dietary treatments: Control = conventional diet containing solvent-extracted defatted soybean meal and corn; EENO = diet containing extruded expelled defatted normal-oleic soybean meal and corn; FFNO = diet containing full-fat normal-oleic soybean meal and corn; FFHO = diet containing full-fat high-oleic soybean meal and corn. Forty-eight White Shaver hens were individually housed and randomly assigned to one of four isocaloric, isonitrogenous dietary treatments (12 replicate cages/treatment) and provided feed and water freely for 21 days. At termination, 12 eggs were collected from each treatment group for chemical analysis by an AOAC-approved commercial laboratory using AOAC-approved methods. ^2^ Crude Fat content = g crude fat/g total sample weight × 100. Fatty acid content = g of fatty acid/g total lipid content × 100. *p*-value = statistically significant differences *p* < 0.05 by analysis of variance (ANOVA). ^a–d^ Means within the same row lacking a common superscript differ significantly (*p* < 0.05).

## Data Availability

The data presented in this study are available on request from the corresponding author.

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
