# Peer review of "Full-Fat Soybean Meals as an Alternative Poultry Feed Ingredient—Feed Processing Methods and Utilization—Review and Perspective"

_animals, 2024, doi:10.3390/ani14162366_

Round 1

Reviewer 1 Report

Comments and Suggestions for Authors

This review has examined the current feed processing methods and utilization of conventional and full-fat high-oleic soybeans as preferable feed stock rations for poultry. The topic is interesting. The paper was well written. The following revision could improve the quality of the paper.

The necessary of using the FFSBM in feed need to be simply explained the abstract and introduction section.

L26-27, what is the crude fat in the FFSBM?

Figure 1, please reorganized the picture and makes it simple and easier to read and understand.

L375, please using the right way to display the footnote and references.

Please using the standard way of three lines to present the all the tables.

L437, please check if it appropriate to present/cite the results from the other publication? It is better to combined and summarized the similar trials.

Table 3, why the weight gain of the birds was reduced with the increasing the levels of FFSBM?

L485-489, please check the IFA and use the right way to express the footnotes.

Please add the replicates n=? in the footnote. All the P value need to be writing in italic. The space needs to be put before and after ‘<’ and etc. Please check throughout the paper.

Author Response

Academic Editor Notes: The following revision could improve the quality of the paper.
Author Response: Thank you for your constructive and helpful comments. All requested edits have been completed per the reviewer’s comments.

Comments: The necessary of using the FFSBM in feed need to be simply explained the abstract and introduction section.

Author Response: Respected reviewer, we have included further text regarding the importance and necessity of FFSBM in the abstract and introduction. (Lines 27-30 in Abstract and line 69-85 in Introduction).

Comments: L26-27, what is the crude fat in the FFSBM?

Author Response: In FFSBM, it is around 18 to 20% and added in the text in line 29 of the abstract.

Comments: Figure 1, please reorganized the picture and makes it simple and easier to read and understand.

Author Response: Thanks, a new picture has been added to simplify the process further.

Comments: L375, please using the right way to display the footnote and references.

Author Response: We have updated the footnote and references in all table.

Comments: Please using the standard way of three lines to present the all the tables.

Author Response: All the tables have been updated accordingly.

Comments: L437, please check if it appropriate to present/cite the results from the other publication? It is better to combined and summarized the similar trials.

Author Response: We did not present the whole table from the original research, and it was only a portion of the actual table with cited reference to give credit to respective author. In different studies, researchers used different levels of FFSBM and it will be difficult to combine different levels into one table. Furthermore, combining all the tables will make it harder for the reader to understand the results. Table has been moved to Line 426.

Comments: Table 3, why the weight gain of the birds was reduced with the increasing the levels of FFSBM?

Author Response: Weight gain was reduced due to a decrease in feed intake. A decrease in feed intake can be due to the presence of anti-nutritional factors in the FFSBM as a result of improper processing or heat treatment. Explanation has been provided in the original text (line 326 – 329)

Comments: L485-489, please check the IFA and use the right way to express the footnotes.

Author Response: We have updated the footnotes in all tables.

Comments: Please add the replicates n=? in the footnote. All the P value need to be writing in italic. The space needs to be put before and after ‘<’ and etc. Please check throughout the paper.

Author Response: Replicate number was added in footnote of table 6 in line 453. Also, we have written the P value in italics, and space has been given before and after “<” throughout the paper.

Reviewer 2 Report

Comments and Suggestions for Authors

Regarding the manuscript entitled Alternative Soy Poultry Feed Ingredients-Feed Processing Methods and Utilization-Review and Perspective. What is the new knowledge the review offers to readers? All information in the review is well-known. The authors should strengthen their review by discussing the perspectives and new methods or alternatives to soybean meal in animal nutrition. In the title what are the alternatives that the authors refer to?

Comments on the Quality of English Language

minor editing

Author Response

Reviewer 2's comments:

What is the new knowledge the review offers to readers?
Author Response: This revised manuscript offers newly published (2023 and 2024) information regarding the effects of the use of full-fat high-oleic soybean meals in poultry diets as compared to conventional commercial defatted soybean meals which are found in tables 7, 8, 9 and 10 of the revised manuscript.

All information in the review is well-known.
Author Response: Information provided within the body of the text pertaining to the use of solvent extracted defatted and extruded expelled defatted soybean meals in poultry diets is well known. However, the information pertaining within the body of the text and tables regarding high-oleic soybeans and full-fat high-oleic soybeans as compared to conventional soy products is not well known comparatively.

The authors should strengthen their review by discussing the perspectives and new methods or alternatives to soybean meal in animal nutrition.
Author Response: We have strengthened the review with discussion in the introduction and perspective sections of the revised manuscript, the effects of the COVID-19 global pandemic on the production and supply of poultry feed components such as solvent extracted defatted soybean meal, palm and coconut oils which directly have affected the price of poultry feed and production cost. Therefore, in this revised review manuscript we aim to discuss the use of full-fat normal-oleic and full-fat high-oleic soybean meals as replacements and alternatives to these vegetable oils and solvent extracted defatted soybean meal in poultry feed formulations. In this review we aim to revisit the use of full-fat soybean meal as a solution and alternative to these current global shortages and rising costs. While full-fat soybean meal prepared from conventional normal-oleic soybean cultivars has existed for over 50 years, utilization of full-fat soybean meal has been under-utilized within the poultry animal feed markets. Moreover, in this review we explore the utilization of high-oleic soybean cultivars for the production of full-fat high-oleic soybean meal is also explored. Thus, in this review we also aim to discuss the feed processing of normal-oleic and high-oleic soybeans with retention of their natural fat content and comparison of these soybean meals.

In the title what are the alternatives that the authors refer to?
Author Response: Per the reviewers helpful comments, we elected to revise the title of this revised manuscript to, “Full-Fat Soybean Meals as an Alternative Poultry Feed Ingredient-Feed Processing Methods and Utilization-Review and Perspective.

Reviewer 3 Report

Comments and Suggestions for Authors

This manuscript focuses more on discussing soybeans and its processing, so it is less appropriate to the objectives and scope of this journal. Furtherore, this manuscript only discusses the condition in the US. Since the title does not indicate conditions for the US, the discussion should be expanded worldwide.

The introduction discusses too much about the history of soybeans but only a little about its processing and utilization. Please provide relevant information according to the title, which can bring the readers to the main topic.Line 57-62           : It is not necessary to write the year in reference, such as 2023a, 2023b, and 2021.

Line 106               : what is mg/gm? Please use the metric unit

Line 138-139      : See comment Line 57-62. This applies also to Lines 211, 301, 312, 322, 417, 439, 455, 475, 485.

Line 345-346      : This sentence should not be separated into different paragraphs.

Line 359               : remove fullstop

Academic Editor Notes

The following revision could improve the quality of the paper.

Comments: Since the title does not indicate conditions for the US, the discussion should be expanded worldwide.

Author Response: Respected reviewer, this paper addresses the soybean and its processing and utilization methods worldwide. However, due credit and acknowledgment were given in the Perspectives section at the end to USDA scientists and related bodies who worked hard to produce different varieties of soybeans.

Comments: The introduction discusses too much about the history of soybeans but only a little about its processing and utilization. Please provide relevant information according to the title, which can bring the readers to the main topic.

Author Response: Respected review, the history of soybeans was discussed only in one page of the introduction to give readers past perspective. However, in the rest of all sections, much emphasis is given to soybean processing, nutrition, and utilization.

Comments: Line 57-62           : It is not necessary to write the year in reference, such as 2023a, 2023b, and 2021.

Author Response: We have addressed and removed the year from references.

Comments: Line 106               : what is mg/gm? Please use the metric unit

Author Response: We updated from mg/gm to g/kg in line 125.

Comments: Line 138-139      : See comment Line 57-62. This applies also to Lines 211, 301, 312, 322, 417, 439, 455, 475, 485.

Author Response: We have removed the year from all references mentioned.

Comments: Line 345-346      : This sentence should not be separated into different paragraphs.

Author Response: This issue has been addressed in the revised review manuscript.

Comments: Line 359               : remove fullstop

Author Response: We have removed the full stop as requested.

Round 2

Reviewer 2 Report

Comments and Suggestions for Authors

Thank you for revisions.

Comments on the Quality of English Language

Minor editing